# The Impact of Antipsychotic Treatment on Neurological Soft Signs in Patients with Predominantly Negative Symptoms of Schizophrenia

**DOI:** 10.3390/biomedicines10112939

**Published:** 2022-11-15

**Authors:** Cristian Petrescu, Ioana R. Papacocea, Crisanda Vilciu, Oana A. Mihalache, Diana M. Vlad, Gabriela Marian, Brindusa E. Focseneanu, Cristian T. Sima, Constantin A. Ciobanu, Sorin Riga, Adela M. Ciobanu

**Affiliations:** 1Neuroscience Department, Discipline of Psychiatry, Faculty of Medicine, ‘Carol Davila’ University of Medicine and Pharmacy, 020021 Bucharest, Romania; 2Department of Psychiatry, ‘Prof. Dr. Alexandru Obregia’ Clinical Hospital of Psychiatry, 041914 Bucharest, Romania; 3Discipline of Physiology, Faculty of Medicine, ‘Carol Davila’ University of Medicine and Pharmacy, 020021 Bucharest, Romania; 4Department of Neurology, ‘Carol Davila’ University of Medicine and Pharmacy, 020021 Bucharest, Romania; 5Neurology Clinic ‘Fundeni‘ Clinical Institute, 022328 Bucharest, Romania; 6Academy of Romanian Scientists‘, 927180 Bucharest, Romania; 7Department of Psychiatry and Psychology, ‘Titu Maiorescu‘ University of Medicine, 040051 Bucharest, Romania; 8Faculty of Medicine, ‘Titu Maiorescu‘ University of Medicine, 040051 Bucharest, Romania; 9Department of Stress Research and Prophylaxis, ‘Prof. Dr. Alexandru Obregia’ Clinical Hospital of Psychiatry, 041914 Bucharest, Romania; 10Romanian Academy of Medical Sciences, 927180 Bucharest, Romania

**Keywords:** antipsychotics, schizophrenia, neurological abnormalities, neurological soft signs

## Abstract

Schizophrenia is a complex and incompletely elucidated pathology that affects sensorimotor function and also produces numerous therapeutic challenges. The aims of this cross-sectional study were to identify the profile of neurological soft signs (NSS) in patients with predominantly negative symptoms of schizophrenia (PNS) compared with patients with schizophrenia who do not present a predominance of negative symptoms (NPNS) and also to objectify the impact of treatment on the neurological function of these patients. Ninety-nine (*n* = 99; 56 females and 43 males) patients diagnosed with schizophrenia according to DSM-V were included; these patients were undergoing antipsychotic (4 typical antipsychotics, 86 atypical antipsychotics, and 9 combinations of two atypical antipsychotics) or anticholinergic treatment (24 out of 99) at the time of evaluation, and the PANSS was used to identify the patients with predominantly negative symptoms (*n* = 39), the Neurological Evaluation Scale (NES) was used for the evaluation of neurological soft signs (NSS), and the SAS was used for the objectification of the extrapyramidal side effects induced by the neuroleptic treatment, which was converted to chlorpromazine equivalents (CPZE). The study’s main finding was that, although the daily dose of CPZE did not represent a statistically significant variable, in terms of neurological soft signs, patients with PNS had higher rates of NSS.

## 1. Introduction

Schizophrenia is a neuropsychiatric disorder with a complex pathophysiology that involves major impairments of thinking, perception, emotions, and behaviour, and it is associated with substantial morbidity, disability, and decreased quality of life [1,2,3,4].

Patients affected by this disorder might exhibit the presence of some neurological abnormalities that are not specific to the disease, but the incidence of these in such patients is greater than in people with other mental illnesses and the normal population. Furthermore, these neurological signs, defined as “soft”, are nonlocalized abnormalities without an exact known relationship to a specific brain lesion and without a clear neurologic diagnostic specificity.

To date, numerous studies [5,6,7,8,9,10,11,12] have concluded that neurological soft signs (NSS) are found in variable proportions in patients with schizophrenia compared with healthy subjects or first-degree relatives and are associated with an early age of onset [6,11], a chronic course of illness [6,11,13,14], negative symptoms [15,16,17], lower IQ, lower education achievements, and a higher score on the Positive and Negative Syndrome Scale (PANSS). The presence of NSS translates into defects in sensory integration (SI), motor coordination (MC), integrative sensory functioning, and complex motor sequencing [18].

Moreover, some authors have advocated the use of NSS scales for the staging of schizophrenia [19,20].

The sensorimotor domain in schizophrenia involves numerous neurological abnormalities that are not limited to neurological disorders caused by adverse reactions to antipsychotic treatments [21,22], as demonstrated by studies on treatment-naïve patients with first-episode schizophrenia who featured more NSS than healthy control subjects [23,24,25,26,27].

Since the majority of authors have agreed on the presence of NSS in patients with schizophrenia, the current desire is to find the substrate of the cerebral damage leading to their presence and to establish treatment guidelines for patients with schizophrenia and NSS [28], especially for cases of treatment resistance, as they tend to have more prominent negative symptoms and a severe course of illness [1].

Regarding the correlation between the presence of NSS in patients with schizophrenia and sociodemographic characteristics, the literature is inconclusive. Bombin et al. [12] determined that although NSS were present in patients with schizophrenia, no correlation could be established between the severity of NSS and the patients’ sex, age, or level of education, in contrast with studies that supported the correlation of NSS with a low education level [27,29,30].

Regarding the substrate of NSS, through progress in neuroimaging, a conclusion that negative symptoms of schizophrenia have common neural substrates in the cerebellothalamic–prefrontal network with NSS has been reached in patients with schizophrenia or first-episode psychosis [31,32,33]. Moreover, imaging studies of brain structure have proposed a prognostic value for a poorer outcome and the presence of predominantly negative symptoms in patients with both brain structural abnormalities and the presence of NSS [34,35].

Key features of schizophrenia include the negative symptoms, which are responsible for a significant proportion of patients’ long-term morbidity and poor functional results [36]. Although described as the most frequent initial symptom of schizophrenia, negative symptoms can occur at any time over the course of the illness. When it comes to the underlying pathophysiology of schizophrenia, the negative symptoms might be the main symptomatology [37]. The European Medicines Agency (EMA) acknowledges negative symptoms as the characteristics of schizophrenia that are currently not properly addressed by existing antipsychotic medications, and these are carefully considered when new drugs are being developed [38,39,40,41]. According to Akinsulore et al. [42], the presence of negative symptoms might predict greater disability in patients with schizophrenia. Multiple studies have found stronger correlations of NSS with cognitive impairment [12], disorganised thinking, working memory deficits [43], or negative symptoms rather than positive symptoms [30,31,44,45,46].

The centrepiece in the treatment of schizophrenia is antipsychotic medication [47]. Positive symptoms are effectively managed by dopamine D_2_ receptor antagonists or partial agonists [37]; nonetheless, negative symptoms generally do not respond to these antipsychotics, and their therapy may necessitate alternate techniques for their management [41]. There is currently no definitive agreement about the impact of these drugs on the severity of NSS. The vast majority of authors support the fact that medication of any type has little to no influence on the existence or evolution of NSS [7,23,24,25,26]. A study [30] that compared patients under treatment with clozapine versus patients treated with conventional neuroleptics concluded that the patients’ NSS scores did not substantially differ between the groups. Another study that used MRI to correlate structural modifications of the brain with NSS in patients with schizophrenia concluded that there was no relationship between cerebellar volumetric measurements and PANSS, neuroleptic dosage, or treatment period, although the total Neurological Evaluation Scale (NES) scores were correlated with marked atrophy in the central white substance of the cerebellum [48]. On the other hand, according to one study, haloperidol therapy tends to cause a drop in NSS [49]. In a 10-week comparative longitudinal study, Buchanan et al. [24] found no differences in NSS between the haloperidol and clozapine treatment groups, except for the scores for the motor coordination items, which decreased in the clozapine group and increased in the haloperidol group. The authors hypothesised that the greater extrapyramidal symptoms (EPS) in the group receiving haloperidol may have contributed to the higher scores for the motor coordination measures. A later study came to the same conclusion that EPS associated with medication might influence the expression of NSS. For the haloperidol (*n* = 37) and risperidone groups (*n* = 19), there were statistically significant associations between the EPS and overall NES score compared with a group of patients treated with clozapine (*n* = 34) [50].

In a study by de Bartolomeis et al. [51], it was determined that higher illness severity, higher antipsychotic doses, and high scores on the NES subscales of sensory integration and other signs were major predictors of treatment-resistant schizophrenia.

NSS are considered more likely to be an intrinsic component of schizophrenia rather than a side effect caused by neuroleptic therapy, as they are present in patients who are receiving neuroleptic medication and in untreated or first-episode schizophrenic patients [50,52,53,54,55,56,57]. Even so, medication-related EPS may have an impact on the expression of NSS [51], as typical or conventional neuroleptic agents may cause Parkinsonian symptoms or akathisia. To clarify this hypothesis, a study by Schröder et al. [58] demonstrated that dopamine receptor D2 (D2R) occupancy in the nigrostriatal dopamine system (the presumed cause of extrapyramidal side effects [59]) was not correlated with NSS. Instead, the single-photon emission computed tomography (SPECT) results showed that the upregulation of the striatal dopamine D2 receptor was significantly correlated with the scores for Parkinsonian side effects but not with NSS. Moreover, a 2010 meta-analysis [60] that included 12 studies regarding Parkinsonism and dyskinesia in antipsychotic-naive schizophrenia patients concluded that dyskinesia and Parkinsonism were found to be substantially correlated with schizophrenia, and the results showed that these movement disturbances were connected to schizophrenia itself and could not be explained on the basis of the use of antipsychotic medication.

To sum up, studies have indicated that NSS are present in a significant proportion of patients with schizophrenia and are not necessarily related to the patients’ age or stage of schizophrenia, information which may be useful in clinical assessments of and research into this pathology [9].

The development of chlorpromazine (CPZ) in the 1950s completely changed how schizophrenia was treated [61]. Later, in the 1960s, clozapine, the first of the atypical antipsychotics, was developed [62]. For many years, it has been the major therapy for treatment-resistant schizophrenia [1].

A 2014 meta-analysis [27] of 17 studies, which also included neuroleptic treatments, showed that the majority of reviewed studies indicated a decline in NSS in individuals receiving any type of neuroleptic medication, but the exact moment when the treatment was initialised was not completely defined in all the studies included in the meta-analysis. In a more recent review [63], numerous studies were taken into consideration by the authors, who showed links between motor coordination and negative symptoms, as well as positive associations over time between overall NSS scores and negative symptoms. Furthermore, the authors implied no strong correlation between the daily dose of antipsychotics and NSS scores.

The present study was aimed to describe the profile of NSS in treated schizophrenia patients and whether NSS have correlations with sociodemographic characteristics, the severity of symptoms, or the daily dosage of the treatment. We also aimed to estimate the frequency of extrapyramidal symptoms and neurological soft signs in a sample of schizophrenic patients and to analyse whether neurological soft signs are more pronounced in patients with predominantly negative symptoms. Another purpose of this study was to examine groups of schizophrenic patients treated with antipsychotics to determine if the presence and/or severity of EPS influenced the expression of NSS.

## 2. Materials and Methods

### 2.1. Setting and Subjects

This cross-sectional study included 99 psychiatric inpatients (56 females and 43 males) recruited from the Prof. Dr. Alexandru Obregia Psychiatry Hospital in Bucharest, with ages ranging from 18 to 64 years and who met the DSM V criteria [64] for schizophrenia. The patients had been on antipsychotic medication for more than three weeks, with a mean daily dose of antipsychotics of 424 mg of chlorpromazine equivalent (CPZE) [65,66,67,68]. The research received approval from the Prof. Dr. Alexandru Obregia Psychiatry Hospital Ethics Committee (approval number 89, 7 June 2022).

All the participants provided written informed consent after the procedures of the study had been fully explained, in accordance with the Declaration of Helsinki and according to the country’s law. The exclusion criteria of the study were as follows: patients who refused to participate in the study or did not provide informed consent; those with mental retardation, an organic brain disorder, a history of substance dependence/abuse as defined by DSM-V [64], a history of severe head trauma, a history of neurological disorders or other severe medical diseases, nonschizophrenia psychotic disorders (including brief psychotic disorder, schizophreniform disorder, schizoaffective disorder, delusional disorder, schizotypal personality disorder, and affective psychosis), and/or a history of other nonpsychiatric drugs with neurological side effects; and those aged outside the 18–65 year range.

As it is considered that negative symptoms have a higher burden of illness and are a valid target for drug development [38,39,40], the included patients were divided into a subgroup of schizophrenia patients with predominantly negative symptoms (PNS) and a subgroup of patients with non-predominantly negative symptoms (NPNS). Regarding the medication, 4 patients were receiving conventional neuroleptic treatment, 86 were atypical neuroleptics, and 9 patients had a combination of 2 atypical neuroleptics. The mean daily dose of biperiden equivalent received for the 24 patients undergoing anticholinergic treatment was 1.54 mg (SD = 0.58)**.** The investigators did not interfere in the choice of neuroleptics or in the given daily dosage.

### 2.2. Measurements

Sociodemographic and medical data were collected from the participants and their families through verbal responses to several questions and from the patients’ medical written or electronic files. They included the patients’ medical history, years of education, marital status, socioeconomic level, psychiatric and medical history, duration of illness, age of onset, age of first hospitalization, number of hospitalizations, and administered treatment.

#### 2.2.1. Assessment of Clinical Symptoms

The clinical symptoms of schizophrenia were assessed with the Positive and Negative Syndrome Scale (PANSS) [69]. Clinical assessments of patients were performed on the same day as their neurological assessments.

As the EMA (European Medicines Agency) [40] guidelines require predominantly negative symptoms in medical trials to study the effect of drugs on negative symptoms in schizophrenia, the following definitions of predominantly negative symptoms were applied:

(1) A baseline score of ≥4 (moderate) on at least 3 or ≥5 (moderately severe) on at least 2 of the 7 negative subscale items and a PANSS positive scale score of less than 19 [70];

(2) A PANSS negative subscale score of ≥ 6 points over the PANSS positive subscale score [71];

(3) A PANSS negative subscale score of at least 21 and at least 1 point greater than the PANSS positive subscale [72].

Furthermore, Rabinowitz et al. 2013 [73] validated this definition in their study.

To correlate the PANSS score with the Clinical Global Impression (CGI) [74] score, the method validated by Leucht et al. [75] was used (Table 1).

#### 2.2.2. Assessment of Neurological Signs

Neurological Evaluation Scale (NES)

The neurological soft signs in each group were assessed with the Neurological Evaluation Scale (NES) [6]. The NES is a structured scale that provides scores in four subscales: motor coordination (MC), sequencing of complex motor acts (SCMA), sensory integration (SI), and a subscale comprising cerebral dominance, short-term memory, unusual eye movements, and primitive reflexes. It encompasses a wide spectrum of neurological symptoms in 26 items. According to its standardised guidelines, each item is assessed on a scale of 0 to 2 (0 being typical, 1 being a little disruptive, and 2 being significantly disruptive). The total score and the scores for each of the four subscales were used to assess the degree of neurological impairment.

Simpson–Angus Scale (SAS)

In the patient group, the extrapyramidal side effects of neuroleptics were rated using the Simpson–Angus Scale (SAS) [76]. The Simpson–Angus Scale was devised in 1970 as a tool for evaluating drug-induced Parkinsonism (DIP) and its associated extrapyramidal side effects. Ten items make up the scale: one measures gait (hypokinesia), six quantify rigidity, and another three measure glabellar tap sign, tremor, and salivation. The scoring system for each item is a five-point scale (0–4). The sum of the individual items is divided by 10 to determine the final score. Extrapyramidal symptoms are indicated by a total score over 0.3 [77].

### 2.3. Statistical Analysis and Data Evaluation

For the statistical analysis, *R software* was used with the following packages: getsummary, Table 1, and leaps (Table 2).

## 3. Results

The clinical and demographic parameters used in our study served as the independent variables in a simple/multiple univariate linear regression, with the dependent variable being the total NES score (Table 3).

A paired two-sample *t*-test was performed for the means, and the results are presented in Table 4.

A simple/multiple univariate linear regression was used, with the dependent variable being the SAS score and the independent variables being the demographic, clinical, and paraclinical variables followed in our study, to determine whether there are any predictors for the severity of hard neurological syndromes (quantified by the SAS scores).

The results of the simple univariate linear regression are presented in Table 5.

Multiple univariate linear regression was carried out with the forward selection of the final model. All predictors in the model with statistical significance are listed in Table 6.

The Pearson’s correlation matrix for the NES, daily dose of CPZE, and SAS is given in Table 7.

All the NES subscales and the total NES score had a positive correlation with the SAS score, but almost no correlation was found between the total NES score and the daily dose of CPZE.

For the whole sample, we found weak to moderate positive correlations between all the subscales and the total NES score and a moderately positive correlation between the daily dose of CPZE and the SAS score. Almost no statistical correlation using Pearson’s correlation coefficient (r) was noted between the daily dose of CPZE and the total NES score. A negligible positive correlation (r = 0.15) was obtained between the CPZE and the NES-MC and NES-SCMA subscales (Appendix A). Out of the total 99 participants, 91 (91.92%) exhibited extrapyramidal symptoms, and 7 (7.69%) of those participants had extrapyramidal symptoms regarded as “clinically significant degree of movement disorder” according to the Simpson–Angus Scale (SAS). Regarding the presence of NSS, 71 patients presented abnormalities included in the sensory integration (SI) subscale of the NES, 83 patients presented abnormalities in the motor coordination (MC) subscale, 94 patients presented abnormalities in the sequencing of complex motor acts (SCMA) subscale, and 88 presented abnormalities in the “Other” subscale. Out of the total patients, 98 presented with at least one neurological abnormality scored using the NES.

A comparison of predominantly negative PANSS patients with the remainder is shown in Table 8.

The Pearson correlation matrix for PNS is given in Table 9.

After using Pearson’s correlation for the subgroup of PNS (Table 9), we found the following correlations for the SAS: a weak positive correlation with the patients’ age, duration of illness, number of hospitalizations, and total period of hospitalisation. Regarding correlations with the NES, a weak correlation was found for the sensory integration and the sequencing of complex motor acts subscales and the total value of NES. A moderate correlation (r = 0.69) was found with the daily dose of CPZE.

Regarding the NES, the total NES score had a weak to moderate correlation with the DOI (of all the subscales, “other” had the highest positive correlation, namely, r = 0.53), number of hospitalizations, and total period of hospitalisation. In terms of the daily dose of CPZE, the highest correlation was with the sequence of complex motor acts NES subscale (r = 0.24).

Scores for negative PANSS symptoms were negatively correlated with age of onset and years of education. A positive correlation was observed with the sensory integration and motor coordination subscales of the NES and with the NES total score.

## 4. Discussion

Clinicians should be cautious in diagnosing dyskinesia in all patients with neuroleptic treatment considering that certain neurological signs are an intrinsic part of a disease and not necessarily an adverse reaction to treatment. This research highlights the significance of evaluating all movement abnormalities before beginning antipsychotic medication.

The total NES score was correlated with males (it was higher in males, with an average increase of 3.3 points in the total NES), thus strengthening other previous reports [78], and age (an additional year was found to be associated with a 0.19 point increase in the total NES score), as shown in Table 3. We found no evidence of a correlation between the total NES score and years of education but a strong correlation between the total NES score and retired patients. This was probably due to the disability caused by the disease that required the retirement of these patients (Appendix A).

The results showed a moderate correlation with the age of onset of schizophrenia (1 year more being associated with an increase of 0.23 points in the NES) and a high correlation with the duration of illness (1 year more being associated with an increase of 0.25 points in the NES), which are comparable with the results of other previous studies [6,14,43].

The results also showed a low correlation with age at first treatment and age of first hospitalisation (an increase in age of 1 year being associated with an increase of 0.54 points in the NES) but a strong correlation with the total number of hospitalisations (another hospitalisation being associated with an increase of 0.54 points in the NES) and the total time spent in the hospital (another month of hospitalisation was associated with an increase of 0.80 points in the NES).

We found no correlation with the daily dose of CPZE, which was in line with a study by Herold CJ et al. [79], in which the authors also found no correlation between the total NES score and the daily dose of CPZE of 80 chronic and sub-chronic patients with schizophrenia. Moreover, there was no correlation associated with the lack of the administration of anticholinergic treatment. A strong correlation with left-handed patients was observed.

### 4.1. Correlations with Extrapyramidal Side Effects Documented with the SAS

We found no correlation of the SAS with the age of onset, duration of illness, age at first treatment, or age at first hospitalisation, but we found a strong correlation with total hospitalisations and the amount of time spent in hospital (for the number of hospitalisations, an increase by one hospitalisation was associated with an increase of 0.15 in the SAS; for the cumulative duration of hospitalisations, an increase of 1 month was associated with an increase of 0.21 in the SAS score), as presented in Table 5. These results are similar to those of another study [80] and could be explained by the fact that a long period of time spent in the hospital is caused by a more severe symptomatology, which probably requires higher doses of treatment that produce extrapyramidal adverse reactions. Regarding the correlations between extrapyramidal side effects and treatment, there was a strong correlation, where an increase of 100 mg in the daily dose of CPZE was associated with an increase of 1 point in the SAS; moreover, a moderate correlation was found for the association of the SAS with anticholinergic medication (its absence being related to a decrease of 1.2 points in the SAS), which is similar to the results of another study [81] and a 2009 meta-analysis [82]. Moreover, anticholinergic medications have also been linked in other studies to cognitive impairments in people with schizophrenia [83,84].

Regarding the type of treatment, when we compared treatment with typical antipsychotics with atypical or two atypical antipsychotics, we observed that the administration of atypical antipsychotics was associated with a decrease of 3.9 points in the SAS and the administration of two atypical antipsychotics was associated with a decrease of 3.6 points in the SAS.

### 4.2. Patients with Predominantly Negative Symptoms (PNS) (n = 39) vs. Patients with Non-Predominantly Negative Symptoms NPNS (n = 60)

There was no statistical correlation between the two subgroups regarding the years of education, age of onset, duration of illness, age at first treatment, number of hospitalizations, or total period spent in hospital. Overall, as shown by Table 8, the patients with predominantly negative symptoms (*n* = 39) had almost the same years of education as the NPNS patients (mean (SD) = 12.38 (1.90) vs. 12.42 (1.99), respectively). For the daily dose of CPZE, there was no statistically significant difference between the two subgroups (*p* = 0.38; for CPZE (mg), mean (SD) = 446.79 (179.84) for PNS patients vs. mean (SD) = 409.58 (241.99) for NPNS patients).

We found no correlation between the total NES score and the total PANSS score of PNS patients or between the total PANSS score of NPNS patients with the “Other” subscale of the NES. Moderate evidence of a correlation between the total PANSS score and the sensory integration, motor coordination, and sequencing of complex motor acts NES subscales in patients with PNS compared with those with NPNS was found. Additionally, there was a strong statistical correlation between the total PANSS score and the total NES score in the PNS vs. NPNS subgroups. These results are in line with previous reports [12,51,85,86] but are in contrast to others [9], in which the authors found weak correlations between the NES and PANSS scores, especially with the negative symptoms, and concluded that the NES score is a variable independent of the PANSS score. A potential explanation for the relationship between NSS and negative symptoms in schizophrenia can be sought at the level of brain architecture. Studies using imaging techniques have shown a relationship between specific brain changes, such as a decreased grey matter volume of the frontotemporal cortex and orbitofrontal cortical thinning, and the presence of negative symptoms in schizophrenia, particularly avolition and apathy, suggesting that these symptoms originate in the prefrontal network [87,88,89,90,91]. On the other hand, other studies that used brain imaging methods, which aimed to identify brain structural changes that correlate with the presence of NSS in patients with schizophrenia or a first psychotic episode, identified structural changes in similar regions [92,93,94] to those correlated with negative symptoms. Future research is thus needed to confirm whether there are certain areas or common structural changes in the brain that correlate with both the onset of negative symptoms and the pathogenesis mechanisms of NSS.

The present study had several limitations. First, patients in this study were not first-episode patients, all were undergoing neuroleptic treatment, and some were also on benzodiazepine medication at the time of evaluation. Nevertheless, as we stated previously, according to the specialised literature, neuroleptic treatment does not seem to significantly influence NSS. For studies regarding only the presence of NSS in patients on the schizophrenia spectrum, it is our opinion that it would be ideal to include drug-naïve patients. Secondly, the fact that every individual in the present study had a history of hospitalisation may indicate that the majority of them have had a more severe course of disease, making it unlikely that the present findings may be applied to other generalised contexts.

Another possible limitation of the present study was the use of the PANSS to quantify negative symptoms. The PANSS negative symptoms subscale contains certain items that are no longer considered relevant in the classification of negative symptoms. With the emergence of new scales for the analysis of negative symptoms, such as The Clinical Assessment Interview for Negative Symptoms (CAINS) and The Brief Negative Symptom Scale (BNSS) [95,96], we consider it appropriate to use them in future studies that exclusively target negative symptoms in schizophrenia. However, the present study aimed to analyse not only negative symptoms but also the entire psychopathological symptomatology. In addition, we used criteria to identify those patients with predominantly negative symptoms according to studies that validated this definition by using the PANSS. Moreover, some authors suggest only using the N1, N2, N3, N4, and N6 items when considering analyses of negative symptom by using the PANSS [97]. Additionally, future research of this kind may apply scales, such as the Calgary Depression Scale for Schizophrenia CDSS [98], to better differentiate depressive symptoms because they might affect the ratings for negative symptoms. One more significant constraint was the total number of patients included in the study. Finally, another important limitation was the cross-sectional design of the study, in contrast to a longitudinal one, that allowed us to better observe the impact of the treatment on the psychiatric and neurological symptoms, thus acting as a call for a large, multicentre longitudinal study.

## 5. Conclusions

This study was conducted to demonstrate the presence of neurological soft signs among patients with schizophrenia and to show the correlations between neurological soft signs and the presence of predominantly negative symptoms, treatment, and extrapyramidal side effects. By comparing patients on the basis of how negative their symptoms were in relation to their overall disease status and the presence of NSS, the current analysis was aimed to address the problem of the specificity of the effects of therapy on negative symptoms. Our research demonstrated that after schizophrenia patients were divided into PNS and NPNS subgroups, the most significant correlations were lost in the NPNS patients but preserved and even increased in the PNS patients who, for instance, had higher total NES and PANSS scores, required higher daily doses of antipsychotic drugs, and had longer cumulative hospitalised periods. In conclusion, we found that neurological soft signs were more prevalent in schizophrenic patients with predominantly negative symptoms compared with NPNS patients. This result raises the possibility that PNS patients might have a significant mediating role in the relationships between NSS and the variables examined in the present investigation.

The overall quality of life of persons suffering from schizophrenia is influenced by both the disorder and how it is treated, which results in a variety of adverse effects of antipsychotic drugs. Thus, the goals of future studies should be to improve the quality of life for patients with schizophrenia and to decrease the frequency of negative symptoms by extending the arsenal of psychiatrists with next-generation medication.

The most significant findings in relation to extrapyramidal symptoms were that they were strongly correlated with the daily dose of CPZE, the absence of anticholinergic treatment, and the type of treatment, particularly when two antipsychotics were combined. It is important to note that our current findings do not support the routine use of antipsychotic polypharmacy. Furthermore, at this point, it is impossible to say with certainty that a technique of this kind would never have an acceptable risk–benefit ratio, but more information on the possible effects of combination therapy will undoubtedly arise from future studies.

Regarding patients with predominantly negative symptoms, the present study could not objectify statistically significant differences between these patients and NPNS patients in terms of treatment; instead, an important finding was the fact that patients with PNS presented significantly more NSS than NPNS, thus strengthening the argument that negative symptoms in schizophrenia are distinguished by a unique set of neuropsychiatric features. In addition, compared with the rest of the schizophrenic patients in the present study, these patients displayed higher total scores for the PANSS.

Once again, we want to emphasise the importance of identifying the optimal antipsychotic treatment of the negative symptoms associated with schizophrenia and to reinforce the hypothesis that NSS comprise a trait that rather correlated with negative symptoms. There is a potential issue of subjectivity in the challenge of accurately judging negative symptoms in the presence of rather severe positive symptoms. Thus, it is important to distinguish between the decision to rank or treat a symptom. The issue is not one of operational criteria alone but of whether we view negative symptoms as a physiologically independent characteristic of schizophrenia, with a separate road to functional handicap and a separate opportunity for a particular treatment.

## Figures and Tables

**Table 1 biomedicines-10-02939-t001:** Patient’s symptomatology corelated with CGI scores.

CGI Score	*n*
1 = Normal, not at all ill	0
2 = Borderline mentally ill	2
3 = Mildly ill	13
4 = Moderately ill	24
5 = Markedly ill	45
6 = Severely ill	15
7 = Among the most extremely ill patients	0
Total	99

CGI, Clinical Global Impression; *n,* number of patients.

**Table 2 biomedicines-10-02939-t002:** Descriptive statistical analysis of the studied variables.

Variable	Global (*n* = 99)
Sex	
F	56 (56.6%)
M	43 (43.4%)
Age	
Mean (SD)	30.6 (10.4)
Median (Min, Max)	26.0 (18.0, 65.0)
Environment	
R	12 (12.1%)
U	87 (87.9%)
Years of education	
Mean (SD)	12.4 (1.94)
Median (Min, Max)	12.0 (8.00, 18.0)
Economic status	
Employed	11 (11.1%)
Retired	44 (44.4%)
Unemployed	37 (37.4%)
Student	7 (7.1%)
Age of onset	
Mean (SD)	22.5 (4.67)
Median (Min, Max)	21.0 (17.0, 40.0)
Duration of illness	
Mean (SD)	8.15 (7.78)
Median (Min, Max)	5.00 (1.00, 35.0)
Age at first treatment	
Mean (SD)	22.8 (4.72)
Median (Min, Max)	21.0 (18.0, 40.0)
Missing	3 (3.0%)
Age at first hospitalisation	
Mean (SD)	23.3 (5.24)
Median (Min, Max)	22.0 (18.0, 40.0)
N/A	2 (2.0%)
Number of hospitalisations	
Mean (SD)	5.27 (4.21)
Median [Min, Max]	4.00 (1.00, 25.0)
Cumulative hospitalised period	
Mean (SD)	3.98 (3.05)
Median (Min, Max)	3.50 (0.500, 15.0)
PANSS, CGI correlation	
Mean (SD)	4.59 (0.969)
Median (Min, Max)	5.00 (2.00, 6.00)
PANSS P	
Mean (SD)	21.6 (6.06)
Median (Min, Max)	22.0 (8.00, 35.0)
PANSS N	
Mean (SD)	21.4 (6.32)
Median (Min, Max)	21.0 (8.00, 39.0)
PANSS, general	
Mean (SD)	41.8 (8.66)
Median (Min, Max)	41.0 (20.0, 65.0)
PANSS, total	
Mean (SD)	84.8 (16.8)
Median (Min, Max)	86.0 (42.0, 123)
PANSS, predominantly negative	
Yes	39 (39.4%)
No	60 (60.6%)
Type of treatment	
TA	4 (4.0%)
AA	86 (86.9%)
2 AA	9 (9.1%)
Daily dose of CPZE	
Mean (SD)	424 (219)
Median (Min, Max)	400 (75.0, 1500)
Anticholinergic treatment	
Yes	24 (24.2%)
No	75 (75.8%)
NES, sensory integration	
Mean (SD)	1.67 (1.52)
Median (Min, Max)	2.00 (0, 7.00)
NES, motor coordination	
Mean (SD)	2.08 (1.60)
Median (Min, Max)	2.00 (0, 8.00)
NES, sequencing of complex motor acts	
Mean (SD)	3.12 (1.98)
Median (Min, Max)	3.00 (0, 8.00)
NES, other	
Mean (SD)	3.57 (2.62)
Median (Min, Max)	3.00 (0, 10.0)
NES, total	
Mean (SD)	10.5 (5.50)
Median (Min, Max)	10.0 (0, 22.0)
SAS	
Mean (SD)	3.04 (2.01)
Median (Min, Max)	3.00 (0, 9.00)

F: females; M: males; R: rural; U: urban; PANSS: Positive and Negative Syndrome Scale; CGI: Clinical Global Impression; PANSS P: PANSS positive symptoms; PANSS N: PANSS negative symptoms; NES: Neurological Evaluation Scale; SAS: Simpson–Angus Scale; CPZE: chlorpromazine equivalent; TA: typical antipsychotic; AA: atypical antipsychotic; 2 AA: combination of two atypical antipsychotics.

**Table 3 biomedicines-10-02939-t003:** Simple univariate linear regression.

Predictors	*n*	Beta (95% CI)	*p*-Value
Sex	99		
F		—	
M		3.3 (1.3 to 5.4)	0.002
Age	99	0.19 (0.09 to 0.29)	<0.001
Environment	99		
R		—	
U		−2.5 (−5.8 to 0.78)	0.138
Marital status	99		
With a partner		—	
No partner		0.01 (−2.4 to 2.5)	0.991
Years of education	99	−0.41 (−1.0 to 0.15)	0.157
Economic status	99		
Employed		—	
Retired		6.8 (3.4 to 10)	<0.001
Unemployed		3.3 (−0.11 to 6.7)	0.061
Student		1.9 (−2.9 to 6.7)	0.436
Age of onset	99	0.23 (0.01 to 0.46)	0.048
Duration of illness (years)	99	0.25 (0.12 to 0.38)	<0.001
Age at first treatment	96	0.20 (−0.03 to 0.43)	0.098
Age at first hospitalisation	97	0.20 (−0.01 to 0.40)	0.069
Number of hospitalizations	99	0.54 (0.30 to 0.77)	<0.001
Cumulative hospitalised period (months)	99	0.80 (0.47 to 1.1)	<0.001
Daily dose of CPZE	99	0.001 (−0.003 to 0.007)	0.438
Anticholinergic	99		
Yes		—	
No		−0.45 (−3.0 to 2.1)	0.732
Dominance	99		
L		—	
R		−5.2 (−8.5 to −1.9)	0.003

F: females; M: males; R: rural; U: urban; CPZE: chlorpromazine equivalent; L: left; R: right. Average values, including standard deviations from the total score and the respective subscores of the NES at the time of hospitalisation, are presented.

**Table 4 biomedicines-10-02939-t004:** Paired two-sample *t*-test.

Paired Two-Sample *t*-Test for Means			Paired Two-Sample *t*-Test for Means		
	PANSS negative	Total NES		Total PANSS	Total NES
Mean	21.42	10.45	Mean	84.84	10.45
Variance	39.96	30.27	Variance	281.418	30.27
r	0.33		r	0.19	

PANSS: Positive and Negative Syndrome Scale; NES: Neurological Evaluation Scale; r: Pearson’s correlation coefficient.

**Table 5 biomedicines-10-02939-t005:** Simple univariate linear regression.

Predictors	*n*	Beta (95% CI) ^1^	*p*-Value
Sex	99		
F		—	
M		0.75 (−0.04 to 1.5)	0.066
Age	99	0.02 (−0.02 to 0.06)	0.385
Environment	99		
R		—	
U		−0.71 (−1.9 to 0.50)	0.253
Marital status	99		
With a partner		—	
No partner		0.21 (−0.69 to 1.1)	0.649
Years of education	99	−0.19 (−0.39 to 0.01)	0.072
Economic status	99		
Employed		—	
Retired		1.0 (−0.37 to 2.3)	0.16
Unemployed		0.24 (−1.1 to 1.6)	0.729
Student		−0.26 (−2.2 to 1.6)	0.789
Age at onset	99	−0.03 (−0.11 to 0.06)	0.544
Duration of illness (years)	99	0.04 (−0.01 to 0.09)	0.127
Age at first treatment	96	−0.02 (−0.10 to 0.06)	0.645
Age at first hospitalisation	97	0.00 (−0.08 to 0.08)	0.983
Number of hospitalizations	99	0.15 (0.06 to 0.24)	0.001
Cumulative hospitalised period (months)	99	0.21 (0.09 to 0.34)	0.001
Type of treatment	99		
TA		—	
AA		−3.9 (−5.8 to −2.0)	<0.001
2 AA		−3.6 (−5.9 to −1.4)	0.002
Daily dose of CPZE	99	0.01 (0.00 to 0.01)	<0.001
Anticholinergic	99		
Yes		—	
No		−1.2 (−2.1 to −0.31)	0.010
Dominance	99		
L		—	
R		0.05 (−1.2 to 1.3)	0.944

F: females; M: males; R: rural; U: urban; CPZE: chlorpromazine equivalent; TA: typical antipsychotic; AA: atypical antipsychotic; 2 AA: combination of two atypical antipsychotics. ^1 ^CI = confidence interval.

**Table 6 biomedicines-10-02939-t006:** Multiple univariate linear regression forward selection of the final model.

Predictors	Beta (95% CI) ^1^	*p*-Value
Daily dose of CPZE	0.004 (0.003 to 0.006)	<0.001
Number of hospitalizations	0.16 (0.09 to 0.23)	<0.001
Type of treatment		
TA	—	
AA	−2.1 (−3.8 to −0.51)	0.011
2 AA	−2.7 (−4.6 to −0.89)	0.004

CPZE: chlorpromazine equivalent; TA: typical antipsychotic; AA: atypical antipsychotic; 2 AA: combination of two atypical antipsychotics. ^1 ^CI = confidence interval.

**Table 7 biomedicines-10-02939-t007:** Pearson’s correlation matrix for the NES, daily dose of CPZE, and SAS.

	NES-SI	NES-MC	NES-SCMA	NES-Others	NES-Total	SAS	CLPZE MG
NES-SI	1						
NES-MC	0.34	1					
NES-SCMA	0.39	0.34	1				
NES-Other	0.40	0.25	0.28	1			
NES-Total	0.71	0.63	0.7	0.76	1		
SAS	0.33	0.17	0.27	0.26	0.37	1	
CPZE	0.03	0.15	0.15	−0.07	0.07	0.57	1

NES: Neurological Evaluation Scale; NES-SI: NES sensory integration; NES-MC: NES motor coordination; NES-SCMA: NES sequencing of complex motor acts; SAS: Simpson–Angus Scale; CPZE: chlorpromazine equivalent (mg).

**Table 8 biomedicines-10-02939-t008:** Patients with predominantly negative symptoms (PNS) vs. non-predominantly negative symptoms (NPNS).

Variables	PNS *n* = 39	NPNS *n* = 60	*p*-Value ^1^
Ex, n (%)			<0.001
F	13 (33)	43 (72)	
M	26 (67)	17 (28)	
Age, mean (SD)	29.31 (8.72)	31.47 (11.30)	0.29
Environment, n (%)			>0.99
R	5 (13)	7 (12)	
U	34 (87)	53 (88)	
Marital status, n (%)			0.093
With a partner (actual or historical)	7 (18)	20 (33)	
No partner	32 (82)	40 (67)	
Years of schooling (number of years of education), mean (SD)	12.38 (1.90)	12.42 (1.99)	0.94
Economic status, n (%)			0.27
Employed	2 (5.1)	9 (15)	
Retired	16 (41)	28 (47)	
Unemployed	17 (44)	20 (33)	
Student	4 (10)	3 (5.0)	
Age at onset, mean (SD)	21.85 (3.62)	22.87 (5.23)	0.25
Duration of illness (years), mean (SD)	7.46 (7.22)	8.60 (8.15)	0.47
Age at first treatment, mean (SD)	22.08 (3.73)	23.29 (5.25)	0.19
N/A	1	2	
Age at first hospitalisation, mean (SD)	22.67 (4.37)	23.72 (5.75)	0.31
N/A	0	2	
Number of pre-evaluation (pre-diagnosis) hospitalizations, mean (SD)	5.87 (4.73)	4.88 (3.83)	0.28
Cumulative hospitalised period, mean (SD)	4.63 (3.73)	3.56 (2.45)	0.12
PANSS–CGI correlation, mean (SD)	4.92 (0.70)	4.37 (1.07)	0.002
General PANSS, mean (SD)	43.79 (8.08)	40.48 (8.84)	0.058
PANSS Negative, Mean (SD)	26.15 (4.25)	18.35 (5.51)	<0.001
PANSS Positive, Mean (SD)	20.44 (4.91)	22.42 (6.63)	0.091
Total PANSS, mean (SD)	90.38 (13.90)	81.25 (17.60)	0.005
Daily dose of CPZE, mean (SD)	446.79 (179.84)	409.58 (241.99)	0.38
SAS mean (SD)	2.98 (1.97)	3.13 (2.10)	0.73
Anticholinergic, n (%)			0.83
Yes	9 (23)	15 (25)	
No	30 (77)	45 (75)	
Dominance, n (%)			0.11
L	7 (18)	4 (6.7)	
R	32 (82)	56 (93)	
NES—motor coordination, mean (SD)	2.59 (1.71)	1.75 (1.43)	0.013
NES—sensory integration, mean (SD)	2.13 (1.82)	1.37 (1.21)	0.024
NES—sequencing of complex motor acts, mean (SD)	3.67 (2.07)	2.77 (1.84)	0.03
Other NES, mean (SD)	4.05 (2.66)	3.25 (2.57)	0.14
Total NES, mean (SD)	12.49 (5.35)	9.13 (5.23)	0.003

F: females; M: males; R: rural; U: urban; PANSS: Positive and Negative Syndrome Scale; PANSS P: positive PANSS symptoms; PANSS N: negative PANSS symptoms; NES: Neurological Evaluation Scale; CPZE: chlorpromazine equivalent; CGI: Clinical Global Impression. ^1 ^Pearson’s chi-squared test, Welch’s two-sample *t*-test, and Fisher’s exact test.

**Table 9 biomedicines-10-02939-t009:** Pearson correlation matrix for PNS.

	Age	YOE	AAO	DOI	AFT	AFH	NOH	CHP	PANSS P	PANSS N	PANSS, general	PANSS, total	CPZE	NES-SI	NES-MC	NES-SCMA	NES, other	NES, total	SAS
Age	1																		
YOE	−0.02	1																	
AAO	0.59	0.16	1																
DOI (years)	0.91	−0.1	0.2	1															
AFT (years)	0.57	0.14	0.97	0.18	1														
AFH (years)	0.65	0.13	0.89	0.33	0.98	1													
NOH	0.54	0	0.11	0.6	0.05	0.27	1												
CHP	0.44	−0.1	0.03	0.51	−0.01	0.17	0.92	1											
PANSS P	−0.32	−0.22	−0.21	−0.28	−0.25	−0.24	−0.09	0.01	1										
PANSS N	−0.37	−0.35	−0.19	−0.36	−0.19	−0.14	−0.09	0.09	0.69	1									
PANSS, general	0.2	−0.27	0.09	0.2	0.03	0.04	0.1	0.12	0.46	0.29	1								
PANSS, total	−0.1	−0.35	−0.07	−0.09	−0.12	−0.1	0.02	0.11	0.83	0.71	0.83	1							
DD CPZE	0.1	−0.03	−0.08	0.17	−0.06	0.04	0.28	0.3	−0.13	0.13	−0.11	−0.07	1						
NES-SI	0.17	−0.22	0.18	0.11	0.17	0.24	0.34	0.33	0.13	0.3	−0.04	0.11	0.09	1					
NES-MC	0.02	0.04	0.11	−0.02	0.03	0.12	0.12	0.12	0.09	0.2	0.1	0.15	0.08	0.31	1				
NES-SCMA	0.2	−0.1	0.11	0.19	0.14	0.07	0.27	0.38	0.01	0.09	0.31	0.21	0.24	0.22	0.16	1			
NES, other	0.54	−0.21	0.24	0.53	0.27	0.26	0.25	0.23	0.06	0.01	0.27	0.18	0.02	0.28	0.07	0.22	1		
NES, total	0.43	−0.18	0.27	0.38	0.27	0.29	0.41	0.42	0.1	0.2	0.28	0.26	0.17	0.67	0.52	0.62	0.7	1	
SAS	0.38	−0.06	0.06	0.43	0.01	0.2	0.41	0.4	−0.12	−0.01	0	−0.05	0.69	0.4	0.13	0.31	0.28	0.44	1

YOE: years of education; AAO: age at onset; DOI: duration of illness; AFT: age at first treatment; AFH: age at first hospitalisation; NOH: number of pre-evaluation (pre-diagnosis) hospitalizations; CHP: cumulative hospitalised period; PANSS: Positive and Negative Syndrome Scale; PANSS P: positive PANSS symptoms; PANSS N: negative PANSS symptoms; NES: Neurological Evaluation Scale; NES-SI: NES sensory integration; NES-MC: NES motor coordination; NES-SCMA:NES sequencing of complex motor acts; SAS: Simpson–Angus Scale.

## Data Availability

All the data reported within the article are available in anonymised form upon request from the qualified investigators. The data presented in this study are available on request from the corresponding author.

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
