# Peer review of "The Impact of Antipsychotic Treatment on Neurological Soft Signs in Patients with Predominantly Negative Symptoms of Schizophrenia"

_biomedicines, 2022, doi:10.3390/biomedicines10112939_

Round 1

Reviewer 1 Report

The article is devoted to the study of the profile of neurological mild symptoms in patients with predominantly negative symptoms of schizophrenia compared with patients with schizophrenia who do not have a predominance of negative symptoms, as well as to the study of the effect of treatment on the neurological function of these patients. Schizophrenia is a complex, not fully understood pathology that affects sensorimotor function and leads to numerous therapeutic problems. So, this article has both scientific and practical significance.

Unfortunately, the authors used mostly outdated references, only 10% are in the period 2017-2022. I consider this important, since new studies on this topic could now appear.

1. The methodology is appropriate, does not need improvement. Additional control is not needed. 2. The conclusions are consistent with the evidence and arguments presented and answer the main question posed. 3. References are relevant, but most of them are not included in the 5-year period 2017-2022, therefore, are outdated. We need new references on this topic, modern ones. 4. Tables and figures are satisfactory.

Author Response

Dear reviewer,
We appreciate your very important and well-reasoned comments and ideas. We believe that the reports you provided illustrate vital components of the manuscript that we worked hard to enhance.
Additionally, we would like to express our gratitude for your time and the privilege of providing us with these suggestions, which helped to enhance not only this article but also future ones.
In what follows, we will consider answering each point addressed by you for the article
The Impact Of Antipsychotic Treatment On Neurological Soft Signs In Patients With Predominantly Negative Symptoms Of Schizophrenia.
Related to the bibliography, although part of it is outdated, considering the present study to be published in a special commemorative issue, we wanted that any reader can see the evolution of research in the field, from past to present. In addition, due to the natural need to present the most recent information, we have added new bibliographic markers of some works published in the last 5 years, at the suggestion of the reviewers, which we consider to bring new valuable information to the present study and to the specialized literature. Following the thorough revision of the text, we eliminated certain bibliographic markers older than 5 years, which we consider that, although they brought a substantial contribution to the field of research at the time of publication, at this moment there are new studies that bring up-to-date information.
The ultimate aim is our desire to produce an article in accordance with the requirements set forth by Biomedicines, and we remain at your disposal for any suggestions you may have.

Reviewer 2 Report

The authors have submitted a research article regarding an impact of antipsychotic treatment on neurological soft signs (NSS) in patients with schizophrenia by evaluating neurological evaluation scale (NES), illustrating a hypothesis suggesting that, although the not representing a statistically significant outcome, the daily treatment with chlorpromazine could be effective on NSS in patients with predominantly negative symptoms (PNS) rather than those without PNS. The authors discussed the beneficial availability of the antipsychotics and the pharmacologic properties which ameliorate the states of the symptoms, resulting in reliable perspectives. This issue is of interest, and impact of their results is strong. My overall concern with the article describing the current available data regarding beneficial availability of the evaluation of NSS, offer something substantial that helps advance our understanding of effective medicinal management available in clinic.

To strengthen authors’ perspectives, the authors are strongly recommended to add a “toxicology” discussion in detail regarding known antipsychotics’ effect on humans. The opposite, toxicological effects of expected outcomes, if known, may influence largely the authors’ perspective.

Author Response

We appreciate your very important and well-reasoned comments and ideas. We believe that the reports you provided illustrate vital components of the manuscript that we worked hard to enhance.
Additionally, we would like to express our gratitude for your time and the privilege of providing us with these suggestions, which helped to enhance not only this article but also future ones.
In what follows, we will consider answering each point addressed by you for the article
The Impact Of Antipsychotic Treatment On Neurological Soft Signs In Patients With Predominantly Negative Symptoms Of Schizophrenia.
First of all, related to the bibliography, although part of it is outdated, considering the present study to be published in a special commemorative issue, we wanted that any reader can see the evolution of research in the field, from past to present. In addition, due to the natural need to present the most recent information, we have added new bibliographic markers of some works published in the last 5 years, at the suggestion of the reviewers, which we consider to bring new valuable information to the present study and to the specialized literature. Following the thorough revision of the text, we eliminated certain bibliographic markers older than 5 years, which we consider that, although they brought a substantial contribution to the field of research at the time of publication, at this moment there are new studies that bring up-to-date information.
Following your analysis, some spelling errors were detected, thus, with the help of the MDPI English Language Editing, we hope that these mistakes have been corrected. Please find attached the certificate attesting to this fact
Related to the perspective of discussing the toxicology of antipsychotics, we consider it a very good observation since some antipsychotics may cause toxicity by blocking potassium or sodium channels, as well as causing toxicity via alpha-1 adrenergic blockade and anticholinergic effects resulting in tachycardia, hypotension, or prolongation of the QTc interval. However, in the present study, there was no intervention in the decision of patient treatment, this being an observational study. No patient included in the present study presented severe, life-threatening adverse reactions, nor were doses used outside the limits imposed by the guidelines. In our clinic, patient safety comes first, and we respect all good clinical practice guidelines and principles, and all the measures taken, including the presence of extrapyramidal symptoms, where applicable, were reported to the department coordinators. In the closing of this point, we agree with the reviewer's perspective, as the toxicological effects of antipsychotics may influence the perspective of the present study, and we are open to address this issue in our future research.
Finally, we hope that we have addressed all the reviewers' comments and that the revised manuscript format represents the outcome of these. The ultimate aim is our desire to produce an article in accordance with the requirements set forth by Biomedicines, and we remain at your disposal for any suggestions you may have.

Reviewer 3 Report

The authors investigated the profile of neurological soft signs (NSS) in patients with predominantly negative symptoms of schizophrenia (PNS) compared to patients with schizophrenia who do not present a predominance of negative symptoms (NPNS), and also to objectify the impact of the treatment on the neurological function of these patients. The authors found that male, age, retired status, age of onset, duration of illness, total number of hospitalizations, and total time spent in the hospital were predictors of NES score of the patients and the number of hospitalizations, cumulative duration of hospitalizations, daily dose of CPZE, presence of anticholinergic medication, and the type of antipsychotic treatment were predictors of SAS score of the patients.

The paper has the potential to contribute to the existing scientific literature on the NSS of schizophrenia. I have a few comments to further improve the quality of the authors’ paper. I have outlined these issues below:

1.      In Table 8. Patients with predominant negative symptoms (PNS) vs. non - predominant negative 284 symptoms (NPNS).

Duration of illness (months/years??)   months?  or  years?? Please specify.

PANSS positive and negative subscale scores are missing

2.      Table 9. Pearson correlation matrix for PNS.

Please justify why the statistical correction for multiple testing is not used.

3.      Depressive symptoms also influence the presentation of negative symptoms and should be controlled when PNS patients were compared to those with NPNS. For example, Calgary Depression Scale for Schizophrenia (CDSS) can be used to evaluate the severity of Depression. Or this issue should be discussed in the limitation.

Please see:

Galderisi, S., Mucci, A., Buchanan, R.W., Arango, C., 2018. Negative symptoms of schizophrenia: new developments and unanswered research questions. The Lancet Psychiatry 5, 664–677. https://doi.org/10.1016/S2215-0366(18)30050-6

Bègue, I., Kaiser, S., Kirschner, M., 2020. Pathophysiology of negative symptom dimensions of schizophrenia – Current developments and implications for treatment. Neurosci. Biobehav. Rev. 116, 74–88. https://doi.org/https://doi.org/10.1016/j.neubiorev.2020.06.004

4.      Several validated and well-established assessment tools, such as the Positive and Negative Syndrome Scale (PANSS), the Scale for the Assessment of Negative Symptoms (SANS), and the 16-item Negative Symptom Assessment (NSA-16) are available to evaluate negative symptoms and track their course over time. In this study, negative symptoms severity was measured by PANSS negative subscale that assesses for blunted affect, emotional withdrawal, poor rapport, passive/apathetic social withdrawal, difficulty in abstract thinking, lack of spontaneity and flow of conversation, and stereotyped thinking. It is known that PANSS negative subscale may be limited by the inclusion of items that are no longer considered relevant to the negative symptom domain (eg, difficulties in abstract and stereotyped thinking, inattentiveness). Newer scales that have been developed to specifically measure negative symptoms include the Clinical Assessment Interview for Negative Symptoms (CAINS), which covers all 5 negative symptom domains, and the Brief Negative SymptomScale (BNSS). This issue may be briefly discussed in the limitation.

5.      Page 2, line 51

The neurological soft signs (NSS) were identified in initial studies in a proportion of 5% in patients with mental disorders and the presence of them translates into defects 52 in sensory integration (SI), motor coordination (MC), integrative sensory functioning, and 53 complex motor sequencing.

The readers may wonder how to identify the 5% of patients. By using what definition? By using NES = Neurological evaluation scale and what is the cut point to define that the patient has neurological soft  signs?

6.      Page 3, line 103

although total NES scores were correlated with marked atrophy in the central white….

Neurological evaluation scale (NES) scale can be mentioned when the abbreviation of NES first shows in the text.

7.      Page 7 , line 243

3.      Result.s

It should be “Results”

8.      The authors mentioned clozapine in the introduction for several times.  Treatment-resistant schizophrenia was mentioned as well.  However, these variables were not specifically identified in the results.  

9.      In the introduction, the authors stated that they aimed to estimate the frequency of extrapyramidal symptoms and neurological soft signs in a sample of schizophrenic patients, and to analyse whether neurological soft signs are more pronounced in patients with predominant negative symptoms.

However, the “frequency” of extrapyramidal symptoms and neurological soft signs was not found in the results.

10.   In Figure A7, regression line seems to be curve but not a linear one. Is there any possible reason?

11.  The authors used CPZE for the dose of antipsychotics. Biperiden equivalent can be used for representing the daily dose of anticholinergic medication.  

12.  The structure of discussion needs to be refined or written. Discussion is not just a repeat of results.  

13.  Page 13, line 364 Regarding the NES scores, we found no correlation between PANSS total score of..   

The sentence is not easy to understand and should be rewritten for the readers and      

the results contradictory to previous research should be further discussed.

14.  Conclusion can be more concise.  Page 14, line 394 Clinicians should be cautious in diagnosing dyskinesia in all patients with neurolep-…..

The sentence can be mentioned in the earlier portion of the discussion.

15.  Page 14, line 400.  The most significant correlations were lost in the NPNS patients…..

What correlation, please specify.

16.   Page 14, line 411.   The most significant findings in relation to extrapyramidal symptoms were that they were strongly correlated with the daily dose of CPZE, the absence of anticholinergic treatment, and also with the type of treatment, particularly when two antipsychotics were combined.

The sentence can be mentioned in the earlier portion of the discussion and would be better to briefly discuss.

17.   The article’s main finding was that although the daily dose of CPZE did not represent a statistically significant variable, in terms of exhibiting neurological soft signs, patients with PNS had higher rates of NSS.

The authors may briefly discuss the potential relationship between negative symptoms and NSS. The PANSS negative subscale assesses for blunted affect, emotional withdrawal, poor rapport, passive/apathetic social withdrawal, difficulty in abstract thinking, lack of spontaneity and flow of conversation, and stereotyped thinking. All these items seem to have nothing to do with the dimension of Neurological evaluation scale (NES) including its subdomains of sensory Integration, motor coordination, sequencing of complex motor acts. What are the possible explanations?

In the reviewer’s opinion, the above-mentioned issues need to be addressed by the authors.

Author Response

We appreciate your very important and well-reasoned comments and ideas. We believe that the reports you provided illustrate vital components of the manuscript that we worked hard to enhance.
Additionally, we would like to express our gratitude for your time, and the privilege of providing us with these suggestions which helped to enhance not only this article but also future ones.
In what follows we will consider answering each point addressed by you for the article The Impact Of Antipsychotic Treatment On Neurological Soft Signs In Patients With Predominantly Negative Symptoms Of Schizophrenia.

Regarding point 1, in which the reviewer outlined the fact that PANSS positive and negative subscales are missing in Table 8, we added those values.
Regarding the Pearson correlation matrix for patients with predominantly negative symptoms (PNS) (point 2), it was used as an exploratory technique additionally to  Pearson's Chi-squared test, Welch’s two-sample t-test and Fisher's exact test used in table 8 to further strengthen the results obtained for the PNS patients.
Regarding point 3, we consider it a very good suggestion to use the The Calgary Depression Scale (CDSS) for a better differentiation of negative symptoms, thus we added this aspect to the limitations of the present study, and we will use this suggestion in our future studies.
Related to point 4, namely the use of the The Clinical Assessment Interview for Negative Symptoms (CAINS) and The Brief Negative Symptom Scale (BNSS) scales, we specify the following: although these new generation scales which aim to analyze negative symptoms (blunted affect, asociality, avolition anhedonia, and alogia) according to the Consensus Development Conference on Negative Symptoms, and although the scales have excellent interrater and test–retest reliability as well as strong internal consistency as it was demonstrated by the authors in their studies, the scales are less strongly related to positive symptoms and agitation than PANSS. Furthermore, however, the present study did not only aim to analyze negative symptoms, but the entire psychopathological symptomatology. In addition, we used the criteria to identify those patients with predominantly negative symptoms, according to studies that validated this definition by using the PANSS scale. Although we consider these scales of particular importance in the clinical setting, and we consider them in future studies, unfortunately at present, in our country these scales have not yet been validated. Following your suggestion, we addressed this point in the limitations section.
Regarding point 5, we believe the reviewer made a very valid comment that resulted in the elimination of that paragraph in which was stated that NSS was present in 5% of patients. Being the outcome of early studies on minor neurological symptoms linked to mental illnesses that did not use the NES scale for identification and thus would have confused readers, that paragraph had only a historical importance.
Following your analysis, some spelling errors were detected, thus, with the help of the MDPI English Language Editing, we hope that these mistakes have been corrected. Please find attached the certificate attesting to this fact (points 6 and 7).
Regarding point 8, we mentioned clozapine as being used in the treatment of resistant schizophrenia, but since the present study was not based on the analysis of a certain antipsychotic in relation to the symptoms and the presence of NSS, but of the classes of antipsychotics, it did not analyze in detail either the resistance of certain patients to treatment. These variables mentioned by the reviewer were not taken into account in the present study, but it will be in possible future studies based on resistance to the administered psychotropic treatment. Out of the total participants, nine (9) of them were undergoing treatment with Clozapine at the time of evaluation, thus not being a homogeneous group in terms of the administered molecule, so we used the conversion to Clorpromazine equivalent (CPZE).
In relation to point 9, the reviewer noted that although this study attempted to assess the prevalence of extrapyramidal symptoms and the frequency of neurological soft signs in our sample, the results did not take these factors into account. Therefore, we come with the following response that was added to the results section: out of the total 99 participants, 91 (91.92%) showed extrapyramidal symptoms, of which 7 (7.69%) patients had extrapyramidal symptoms regarded as "clinically significant degree of movement disorder" according to Simpson-Angus Scale (SAS). Regarding the presence of NSS, abnormalities found on the sensory integration (SI) subscale of NES, motor coordination (MC), sequencing of complex motor acts (SCMA), and "Others" subscale of NES were abnormal in 71 patients, 83 patients, 94 patients, and 88 patients, respectively. 98 patients out of the total were found to have at least one neurological anomaly, as determined by the NES total score.
Regarding point 10, the reviewer pointed out that the regression line is curved in figure A7, and enquired about the reason of this. That figure was plotted as part of our “work in progress” during statistical analysis, so our readers can see the steps taken in that procedure. The graph presents a curved regression line as the logarithmic trendline which was used, because our data (NES total score and age), although had a linear positive trendline, the data showed an initial rapid increase followed by a leveling out of NES score as age increases. A drawback of this graph is the fact that we had more patients in the 18-30 years range than in 30-50 years range. In a future longitudinal study, we hope to come up with new data regarding the evolution of NES in time, as there is still a debate in the academic community, with some authors suggesting that such a graph of evolution of NES by age will present in a “U” shape.
In relation to point 11, at the suggestion of the reviewer, for a better highlighting of the data collected about the administered anticholinergic treatment, he suggested that we should report the daily dose of administered anticholinergic in Biperiden equivalent. Following this suggestion, we add the following: “the mean daily dose of Biperiden equivalent received for the 24 patients undergoing anticholinergic treatment was 1.54 mg (SD = 0.58)”.This row can be found added in the structure of the article. In our study, the anticholinergic that was mainly administered was trihexyphenidyl.
Regarding point 12 the reviewer outlined the need of refining the structure of discussion section. We consider this section of our study to present the importance and relevance of our results and also the link between literature studies and our data. Our goal was to write the discussion part with simplicity, clarity, and effectiveness so that it will be shared with our readers in a comprehensible format without missing important results. As we value every aspect of our reviewers’ comments and suggestions, the present manuscript underwent modifications including to this subsection. We hope that these changes are in line with those expected by the reviewers and the editors.  
In relation to point 13, we added the explanation for the contradictory results with another study, related to the correlation between the NES and PANSS scores. In addition, at the reviewer's suggestion, we made changes to that paragraph, hoping for a new form that can be easier to understand by readers.
Regarding point 14 the issue was addressed and the paragraph outlined by the reviewer was mentioned in the earlier part of the discussion.
In relation to point 15, the reviewer emphasized the importance of clarifying which aspects were lost or accentuated following the subdivision into PNS and NPNS. Thus, we have added this clarification that can be found in the text: “our research demonstrates that after schizophrenia patients were divided into PNS and NPNS subgroups, the most significant correlations were lost in the NPNS patients, but they were preserved and even increased in the PNS patients, for instance, they had higher total NES and PANSS scores, required higher daily doses of antipsychotics, and had longer cumulative hospitalized periods”. We would like to point out that following the subdivision, the cumulative duration of hospitalizations increased from an average of 3.98 for the entire group of patients, to an average of 4.63 for the PNS group, the total NES score increased from an average of 10.45 to 12.49 for the PNS and decreased to a mean of 9.13 for the NPNS, and the total PANSS score increased from a group mean of 84.8 to 90.38 for the PNS group. We hope that, following this valuable suggestion, the addition made will better outline the obtained results.
Related to point 16, the reviewer emphasized the need for a discussion related to the following statement: the most significant findings in relation to extrapyramidal symptoms were that they were strongly correlated with the daily dose of CPZE, the absence of anticholinergic treatment, and also with the type of treatment, particularly when two antipsychotics were combined. Regarding the correlation between the absence of anticholinergic treatment and the lower SAS score, a possible explanation is the fact that the majority of patients who did not require anticholinergic treatment, were in earlier stages of the disease, and were receiving antipsychotic treatment for less time, thus not appearing extrapyramidal adverse reactions. At this moment we cannot firmly conclude this aspect. Regarding the correlation between the administration of two antipsychotics and lower SAS scores compared to the administration of a single antipsychotic, the possible explanation for this result is based on the fact that most of the patients who received two antipsychotics, one of which was Clozapine. Being an antipsychotic with more metabolic side effects rather than extrapyramidal side effects, and the second antipsychotic used was in low doses, used for the purpose of augmentation, it is possible that this is the cause of lower scores for extrapyramidal side effects. As far as we know, from the specialized literature, the strategies for increasing the treatment with Clozapine, or another antipsychotic, have contradictory results in terms of adverse reactions. Being a cross-sectional study and an inhomogeneous group in terms of the duration of the disease, we will certainly try to address these aspects in a future longitudinal study. At the same time, in the perspective of a future longitudinal study, it is interesting to answer the question whether this association of antipsychotics leads to the improvement of the NES scores in parallel with the improvement of the PANSS score or not. In summary of this point, we would like to point out that the following paragraph has been added to the text: “it is important to note that our current findings do not support the routine use of antipsychotic polypharmacy. Furthermore, at this point it is impossible to say with certainty that a technique of this kind would never have an acceptable risk-benefit ratio, but undoubtedly, more information on the possible effects of combination therapy will arise from future studies.”

Regarding point 17, the reviewer suggested the need of briefly discussing the potential relationship between negative symptoms and NSS. We believe that the level of cerebral architecture is where the solution most likely resides. Scales that emphasize the existence of both negative symptoms and NSS, point to the effects of these potential brain changes, but only specific imaging tests can definitively prove this. In order to emphasize the importance of future imaging research, we added the following paragraph to the text based on studies that sought to determine the brain structural changes linked to the appearance of either NSS or negative symptoms: “Studies using imaging techniques show a relationship between specific brain changes, such as decreased gray matter volume of frontotemporal cortex and orbitofrontal cortical thinning, and the presence of negative symptoms in schizophrenia, particularly avolition and apathy, suggesting that these symptoms originate in the prefrontal network. On the other hand, studies carried out also by brain imaging methods, which aimed to identify brain structural changes that correlate with the presence of NSS in patients with schizophrenia or first psychotic episode, identified structural changes in similar regions to those correlated with negative symptoms. Future research is thus needed to confirm whether there are certain areas or common structural changes in the brain that correlates with both the onset of negative symptoms and the pathogenesis mechanisms of NSS.”
Finally, we hope that we have addressed all the reviewers' comments and that the revised manuscript format represents the outcome of these. The ultimate aim is our desire to produce an article in accordance with the requirements set forth by Biomedicines, and we remain at your disposal for any suggestions you may have.

Round 2

Reviewer 2 Report

No more comments on this manuscript.

Reviewer 3 Report

The authors address all the reviewer's comments carefully and have improved their manuscript considerably. I have no further comments.